# iProtDNA-SMOTE: Enhancing protein-DNA binding sites prediction through imbalanced graph neural networks

Ruiyan Huang [1], Wangren Qiu[1], Xuan Xiao[1,2], Weizhong Lin [1]*

**1** School of Information Engineering, Jingdezhen Ceramic University, Jingdezhen Jiangxi, China, **2** School of Information Engineering, Jingxi Art & Ceramics Technology Institute, Jingdezhen Jiangxi, China

* linweizhong@jcu.edu.cn

## Abstract

Protein-DNA interactions play a crucial role in cellular biology, essential for maintaining life processes and regulating cellular functions. We propose a method called iProtDNA-SMOTE, which utilizes non-equilibrium graph neural networks along with pre-trained protein language models to predict DNA binding residues. This approach effectively addresses the class imbalance issue in predicting protein-DNA binding sites by leveraging unbalanced graph data, thus enhancing model's generalization and specificity. We trained the model on two datasets, TR646 and TR573, and conducted a series of experiments to evaluate its performance. The model achieved AUC values of 0.850, 0.896, and 0.858 on the independent test datasets TE46, TE129, and TE181, respectively. These results indicate that iProtDNA-SMOTE outperforms existing methods in terms of accuracy and generalization for predicting DNA binding sites, offering reliable and effective predictions to minimize errors. The model has been thoroughly validated for its ability to predict protein-DNA binding sites with high reliability and precision. For the convenience of the scientific community, the benchmark datasets and codes are publicly available at https://github.com/primrosehry/iProtDNA-SMOTE.

## 1. Introduction

Protein-DNA interactions serves as critical regions where transcription factors and other DNA-binding proteins recognize and bind DNA sequences [1] and plays a key roles in life-sustaining processes and cellular functions such as gene expression regulation, DNA replication, and repair [2,3]. Recognizing these binding sites and annotating their functions are essential for revealing gene regulatory networks, identifying disease-related genes, and elucidating mechanisms of drug action [4]. The rapid development of high-throughput sequencing technologies has led to the identification of many protein sequences with unknown functions. However, the identification

**Data availability statement:** All relevant data are within the paper and its Supporting Information files.

**Funding:** This research was funded by the National Natural Science Foundation of China, 62162032 and 32260154, and Technology Projects of the Education Department of Jiangxi Province of China, GJJ2201040 and GJJ2201004.

**Competing interests:** The authors have declared that no competing interests exist.

of these binding sites poses significant challenges for experimental methods duo to the diversity and complexity of protein sequences [5], thereby impeding a comprehensive understanding of biological processes and the discovery of new drug targets and designs [6]. To overcome this, there is a significant scientific and practical need to develop rapid and accurate computational methods for predicting protein-DNA interactions. These methods could provide deeper insights into the mechanisms of these interactions [7], aid in the discovery of novel drug targets, and inform targeted therapeutic strategies [8].

Predicting protein-DNA binding sites involves two main approaches: traditional experimental techniques and computational methods. Traditional experimental methods like protein microarray analysis [9], ChIP-seq [10], x-ray crystallography [11], and Cryo-EM [12]provide valuable data but are costly and complex. In contrast, computational methods process protein sequence data quickly, providing a theoretical foundation for experimental validation and mitigating the limitations of experimental approaches. These computational techniques involve representing protein features based on their sequence, structure, and physicochemical properties, including techniques like one-hot encoding of amino acids [13], PSSM matrices [14], and protein secondary structures [15].

As machine learning and deep learning have advanced, so too has the sophistication of predictive modeling in the field of protein-DNA interactions. Early methods like support vector machines (SVM) [16] and random forests (RF) [17] have been eclipsed by the current generation of deep neural networks [18], which have significantly improved the accuracy and efficiency of predictions. Convolutional networks and graph neural networks (GNN) [19] have been particularly influential in refining the prediction of protein-DNA binding sites. Notably, convolutional networks used by Tayara et al. [20], capsule networks employed by Nguyen et al. [21], and the Inception network utilized by Fang [22] has each yielded substantial improvements in predictive accuracy. Graph neural networks excel at processing graph-structured data, effectively integrating protein sequences, structures, and physicochemical properties to optimize model performance. Yuan et al. introduced the GraphSite model [23], which incorporates tertiary structure information predicted by AlphaFold2. This approach has showcased the potential of these advanced computational techniques in molecular interaction research.

In the domain of bioinformatics, predictive models for protein-DNA binding sites are often impeded by the challenge of imbalanced data distributions. Such imbalances can significantly degrade the models' capacity for generalization and the precision of their predictions. To mitigate these issues, the scientific community has developed an array of sophisticated methodologies, encompassing resampling strategies and ensemble learning techniques. For example, Hu et al. [24] enhanced model efficacy by employing random under-sampling to equilibrate the representation of positive and negative samples, followed by the construction of an ensemble of Support Vector Machine (SVM) classifiers, which were amalgamated via boosting algorithms. Gao et al. [25] innovatively applied multi-instance learning to predict protein-DNA interactions, while Zhu et al. [26] proposed a subsampling strategy

based on the distance of samples from SVM separating hyperplanes, combined with AdaBoost algorithm, to build a protein-DNA binding site predictor that effectively handles data imbalance.

In addressing the challenges posed by imbalanced datasets within the deep learning paradigm, researchers have employed both data-level and algorithmic-level approaches to enhance model performance. For instance, GNN-CL graph neural networks with data interpolation techniques were used for synthesizing new samples to enrich the dataset [27]. The ImGAGN model employed generative adversarial graph networks to generate synthetic minority class nodes thereby optimizing model performance through adversarial processes [28]. The GraphSR model utilized pseudo-labeling techniques to enhance model generalization capabilities [29], while the QTIAH-GNN model introduced a multi-level label perception strategy, alongside parameterized similarity metrics and a specially designed loss function, to balance the predictive emphasis between majority and minority classes [30]. Furthermore, the field has witnessed the emergence of graph convolutional network variants specifically designed to handle imbalanced data, with the introduction of novel loss functions such as Focal Loss [31], which prioritizes the learning from minority class instances. In the domain of imbalanced graph learning [32], researchers have proposed methods such as GraphSMOTE [33], GraphENS [34], GATSMOTE [35], and GraphSHA [36]. These methodologies employ a variety of strategies to strengthen the model's recognition capabilities for minority class nodes and further augment classification performance.

In this study, we introduce the iProtDNA-SMOTE model, an innovative prediction framework that integrates the pre-trained protein language model ESM2 [37] with graph neural network architectures. This model is specifically designed to address the challenge of imbalanced data by leveraging the GraphSMOTE [33] method to enhance recognition of minority class nodes. Furthermore, the iProtDNA-SMOTE model utilizes GraphSage [38] and multi-layer perceptron (MLP) [39] to effectively extract and assimilate sequence-derived features, thereby achieving high-precision prediction of protein-DNA binding sites. Our empirical evaluation across various benchmark datasets substantiates the model has superior predictive accuracy and its unwavering capacity for generalization in predicting DNA binding sites, thereby highlighting its significant potential for application within the biomedical field.

## 2. Materials and methods

### 2.1 Benchmark datasets

In this study, we subjected our iProtDNA-SMOTE method to rigorous evaluation using five reputable datasets that are well-established benchmarks in the field of protein-DNA binding sites prediction. These datasets, designated as TR646 [40], TE46 [40], TR573, TE129, and TE181, represent both training(TR) and testing(TE) components, respectively.

The TR646 dataset comprises 646 DNA-binding protein chains, encompassing a total of 15,636 DNA-binding sites and 298,503 non-binding sites. The TE46 dataset consists of 46 distinct DNA-binding proteins, with 956 DNA-binding sites and 9,911 non-binding sites. Both datasets were introduced through research using the DBPred model, a deep learning approach focused on predicting protein-DNA binding sites from sequence data. Our study employed the TR646 dataset for training, allowing us to explore the intrinsic properties of DNA-binding proteins in depth. The TE46 test set, with a sequence similarity of no more than 30% to the training set, ensures the rigor and independence of our evaluation.

Further, the TR573 dataset consists of 573 DNA-binding protein chains, containing 14,479 DNA-binding residues and 145,404 non-binding residues. The TE129 dataset includes 129 independent DNA-binding proteins, contributing 2,240 DNA-binding residues and 35,275 non-binding residues. These datasets were introduced through research using the GraphBind model, a graph neural network designed to identify nucleic acid binding residues from structural data. The TE181 dataset, introduced through the GraphSite model, includes 181 DNA-binding protein chains with 3,208 DNA-binding residues and 72,050 non-binding residues. This model uses structural insights from AlphaFold2 to classify DNA-binding residues. In our study, the TR573 dataset served for model training, enhancing our understanding of the characteristics of DNA-binding protein. To ensure the independence of the test sets, we restricted a sequence similarity

threshold of no more than 30% between proteins in the TE129 and TE181 datasets and those in the TR573 training set. Moreover, to evaluate the model's generalization capacity, we employed GraphSMOTE technology during model training to refine its handling of imbalanced data.

For the evaluation, we utilized the same data preprocessing procedures as existing models to ensure fairness in assessment. Table 1 presents a comprehensive statistical overview of the four datasets for reference. The TR646 and TE46 datasets were introduced by Patiyal et al. using the DBPred model, while the TR573 and TE129 datasets were presented by Xia et al. based on the GraphBind model. The TE181 dataset was introduced by Yuan et al. through the GraphSite model. We employed TR646 as the training set and its corresponding independent test set, TE46, for evaluation. The TE129 and TE181 test datasets were applied to assess the model trained on TR573. Through the application of these datasets, we comprehensively evaluated the iProtDNA-SMOTE model's capability in predicting protein-DNA binding sites.

## 2.2 The framework of iProtDNA-SMOTE

iProtDNA-SMOTE is a protein-DNA binding site prediction method based on graph neural networks. As illustrated in Fig 1, the iProtDNA-SMOTE process is streamlined into four distinct yet interconnected steps.

1. **Feature Embedding Extraction**: The process begins with extracting feature embeddings using the sophisticated ESM2 large language model. This initial step is pivotal as it translates the raw protein sequences into a high-dimensional space where the underlying biological signals are more pronounced.

2. **Graph Model Construction**: Following embedding extraction, a graph model of the protein sequence is constructed. This graph representation is crucial as it allows the model to consider the spatial relationships between amino acids, which is vital for understanding protein-DNA interactions.

3. **Handling Imbalanced Datasets**: To counteract the common issue of imbalanced datasets, iProtDNA-SMOTE incorporates GraphSMOTE. This technique adeptly adjusts the dataset balance, ensuring that the model does not become biased towards the more frequent classes and enhancing its predictive power across all data points.

4. **Classification of Graph-Structured Data**: Finally, GraphSAGE-MLP is used for the classification of the graph-structured data. This combination of GraphSAGE for neighborhood aggregation and MLP for non-linear classification ensures that the model can accurately predict protein-DNA binding sites.

Each of these steps is designed to work synergistically, providing a comprehensive and robust framework for predicting protein-DNA binding sites with high accuracy and reliability.

**Procedure I: Feature Embedding Extracting.** The amino acid sequence is input into the large language model ESM2, which generates high-dimensional embeddings of size $L \times 2560$, where $L$ represents the sequence length. ESM2, a deep learning model based on the transformer architecture, is specifically designed for understanding and predicting the three-dimensional structure and functions of proteins [41]. Pre-trained on a database containing millions of natural

**Table 1. Summary of benchmark protein–DNA binding datasets.**

| Dataset | | DNA-binding residues | Non-binding residues | % of binding residues |
|---|---|---|---|---|
| Training Dataset | TR646 | 15636 | 298503 | 4.98 |
| | TR573 | 14479 | 145404 | 9.06 |
| Test Dataset | TE46 | 956 | 9911 | 8.87 |
| | TE129 | 2240 | 35275 | 5.97 |
| | TE181 | 3208 | 72050 | 4.26 |

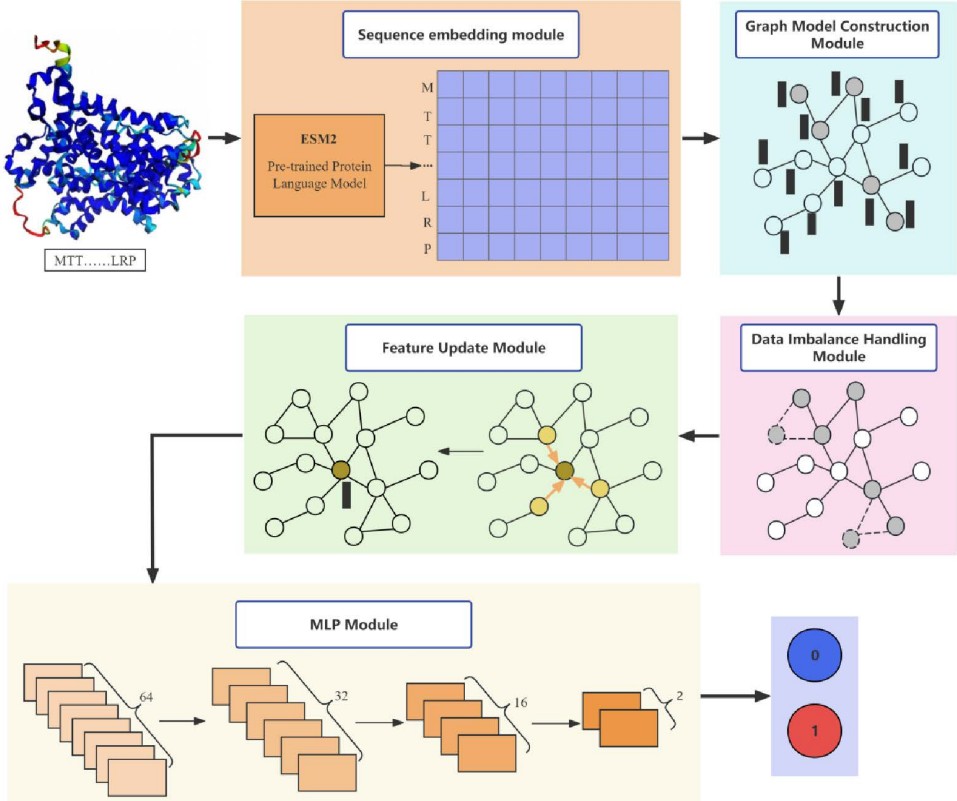

**Fig 1. The workflow of iProtDNA-SMOTE.**

protein sequences, ESM2 is commonly utilized for tasks such as protein structure prediction, functional annotation, and protein-ligand interaction analysis. The feature embeddings produced by ESM2 comprehensively capture information from protein sequences, including chemical properties of amino acid residues, sequence patterns, and interactions between residues [42]. These embedding vectors not only encode information between individual residues but also effectively integrate relationships between residues at different positions within the sequence through the transformer's self-attention mechanism. This integration is crucial as it allows the model to understand the complex interactions within proteins, which are essential for predicting protein-DNA binding sites. It is important to note that in subsequent training steps, the protein feature embeddings generated by ESM2 serve as input data for the graph neural network. These embeddings are generated by the encoder of the ESM2 model, providing precise feature vectors for each amino acid residue in the protein sequence. These feature vectors are utilized as node features in the graph network model, allowing the subsequent graph neural network to effectively process and analyze the protein data for accurate binding site prediction.

**Procedure II: Graph Model Construction.** Utilizing the latest generation protein structure prediction tool developed by DeepMind, AlphaFold3, we obtain precise three-dimensional protein structures. AlphaFold3 represents a significant improvement and expansion over its predecessor, AlphaFold2, with an enhanced evoformer module and diffusion network [43]. Once the protein's three-dimensional structure is acquired, a spatial distance threshold of 8 angstroms is defined. If the distance between the $C_\alpha$ atoms of two residues is less than this threshold, they are connected in the graphical model. This connection facilitates the aggregation of amino acid information that is spatially close, even if it is distant in sequence. Each node in the graph represents an amino acid residue from the 3D structure, with node features derived from the

ESM2 embeddings generated in Procedure I. Every node is labeled accordingly, ensuring that the constructed graphical model comprehensively integrates both the spatial structure and sequence information of the protein.

**Procedure III: Handling Imbalanced Datasets.** The algorithm initiates by pinpointing nodes within the training dataset that belong to minority classes. For each of these minority class nodes, it calculates their similarity to all other nodes across the graph to identify their most proximate neighbors. Subsequently, interpolation techniques are employed to synthesize new nodes. The techniques blend the features of the original minority class nodes with those of their neighbors, thereby generating a fresh set of samples that enrich the minority class representation within the dataset. Crucially, the GraphSMOTE algorithm [33] ensures that the newly minted nodes are not merely isolated additions but are integrated into the graph in manner that reflects realistic relationships and maintains the overall structural properties. This cyclical process of identifying, synthesizing, and integrating new nodes continues until the algorithm achieves the desired numerosity of minority class samples. Through this iterative enhancement, the algorithm not only bolsters the quantity of underrepresented classes but also carefully curates the expansion to safeguard the inherent structure and relational dynamics of the graph.

**Procedure IV: Classification of Graph-Structured Data.** After the aforementioned steps, we proceed by applying GraphSAGE graph convolution operations to the graph structure data. GraphSAGE aggregates neighbor features for each node, effectively mapping them into a new feature space. This process is designed to capture the local structural information of the graph, providing a representation of the graph's topology. The features are subsequently fed into a Multi-Layer Perceptron, which utilizes a series of linear layers with non-linear activation functions to learn complex mappings from features to class labels. This integration of GraphSAGE with an MLP forms the backbone of a comprehensive Graph Neural Network (GNN) model, which is adept at leveraging both the graph's topological structure and the robust learning capabilities of the MLP. Ultimately, the model outputs probability predictions for each node, categorizing them into predefined classes with high accuracy.

## 2.3 Unsupervised protein language models

The architecture of ESM2_t36_3B_UR50D, as shown in Fig 2, accepts a queried amino acid sequence as input and outputs a high-dimensional embedding matrix [44]. This matrix is designed to capable of capture complex biological features and patterns inherent in the sequence. ESM2_t36_3B_UR50D [45] is a variant of the ESM2 model, employing a 36-layer Transformer architecture as its core. It employs self-attention mechanisms that allow each amino acid residue to interact with and learn from others within the sequence, thus understanding their intricate relationships and

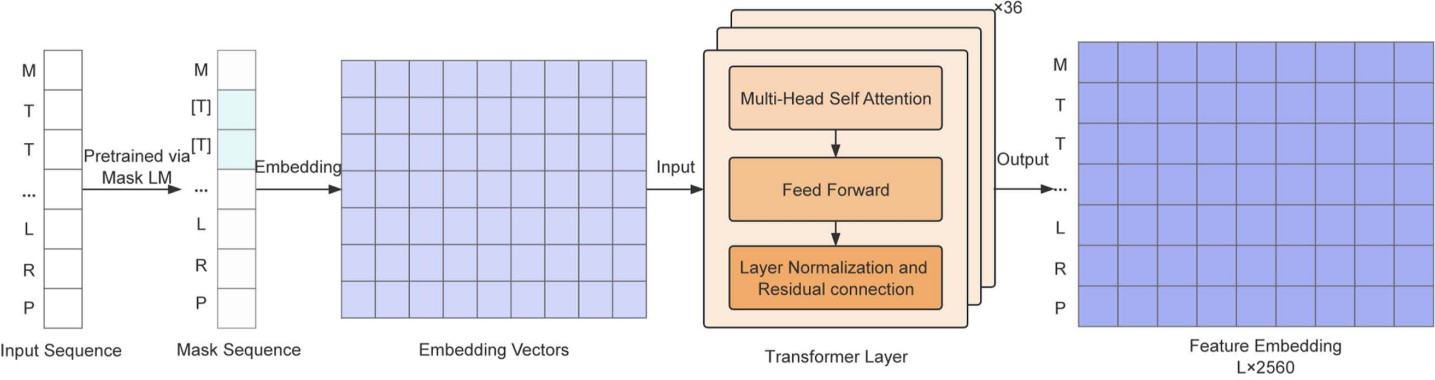

**Fig 2. The workflow of ESM2_t36_3B_UR50D.**

dependencies. This capability is particularly adept at capturing long-range interactions, which are essential for deciphering the three-dimensional structure and function of proteins.

The model incorporates multiple attention heads, each focusing on distinct features of the sequence, collectively generating a comprehensive feature representation. With approximately 3 billion parameters, ESM2_t36_3B_UR50D is trained using masked language modeling objectives to produce feature representations of protein sequences. It benefits from a curated pre-training dataset known as UR50, which comprises over 60 million protein sequences from the UniRef90 database, ensuring a diverse and representative sample of protein sequences.

In summary, the model generates a feature vector of size $L \times 2560$, where $L$ represents the length of the input sequence, and 2560 is the dimensionality of the feature vector, thereby providing a rich, detailed representation of the protein sequence's biological characteristics.

## 2.4 GraphSAGE-MLP network

The GraphSAGE-MLP network, as illustrated in Fig 1, includes a feature update module with SageConv layers, and a MLP module with distinct input, hidden, and output layers.

In the feature aggregation phase, the SageConv layer enhances each node's representation by incorporating features from surrounding nodes. Each node's individual feature, denoted as $h_i$, is combine with the aggregated features of its neighbors, $n_i$, creating an expanded feature vector. This vector is then processed through a linear layer. To introduce complexity, a nonlinear activation function $\sigma$, such as ReLU, is applied to the output of the linear layer, yielding the final feature representation for each node. The mathematical computation is as follows:

$$n_i = \sum_{j \in \mathcal{N}(i)} h_j \tag{1}$$

$$h_i' = [h_i, n_i] \tag{2}$$

$$z_i = W \cdot h_i' + b \tag{3}$$

$$h_i^{(l+1)} = \sigma(z_i) \tag{4}$$

where $n_i$ represents the aggregation result of neighboring features of node $i$, $\mathcal{N}(i)$ is the set of neighbouring nodes of node $i$, and $h_j$ denotes the feature vector of neighbor node $j$. $W$ stands for the weight matrix, and $b$ is the bias term, collectively defining the linear transformation.

The MLP in our model is structured with an input layer, two hidden layers, and an output layer, each linked by nonlinear activation functions. The input layer takes the original features of a node and, through linear transformations, projects them into a higher-dimensional space in the first hidden layer. Here, each neuron calculates a weighted sum of its inputs and includes a bias term. The LeakyReLU function is then applied to introduce nonlinearity.

The output from the first hidden layer, denoted as $a^{(1)}$, is passed to the second hidden layer. It follows a similar process of linear transformation and LeakyReLU activation to further refine the feature representation. The final output layer receives these transformed features and, through a linear transformation, maps them to a space where each dimension represents a class in the classification task. Unlike the hidden layers, the output layer uses the softmax function to convert the output into a probability distribution across the different classes. The computation proceeds as follows:

$$z^{(1)} = w^{(1)}x + b^{(1)} \tag{5}$$

$$a^{(1)} = LeakyReLU\left(z^{(1)}\right) \tag{6}$$

$$z^{(2)} = w^{(2)}a^{(1)} + b^{(2)} \tag{7}$$

$$a^{(2)} = LeakyReLU\left(z^{(2)}\right) \tag{8}$$

$$\hat{y} = softmax\left(z_{out}\right) = \frac{e^{z}out}{\sum_{l=1}^{K} e^{z}l} \tag{9}$$

Where, $x$ represents the input features, $w^{(1)}$ is the weight matrix, and $b^{(1)}$ is the bias vector. $z_{out}$ denotes the linear output of the output layer, $K$ is the total number of classes, and $\hat{y}$ represents the model's predicted class probabilities.

To tackle the issue of class imbalance in binary classification tasks, we utilize the focal loss function. This function adjusts the loss given to correctly classified samples, down weighting those that are easily classified and upweighting those that are challenging to classify correctly. By doing so, it encourages the model to focus more on the samples that are hard to classify accurately. The computation formula is as follows:

$$L = -\alpha(1 - pt)^{\gamma} \log(pt) \tag{10}$$

where $L$ represents the value of the loss function, $\alpha$ is a weight parameter balancing positive and negative samples to adjust for class imbalance, $\gamma$ is an exponent that adjusts the model's focus on easy versus hard samples, reducing the weight of easily classified samples and increasing that of hard-to-classify samples, and pt denotes the model's predicted probability of the positive class.

## 2.5 Evaluation indices

In this study, we utilized six metrics—specificity (Spe), precision (Pre), recall (Rec), F1 score, Matthews correlation coefficient (MCC), and AUC—to evaluate the proposed method, ensuring consistency with previous research. These metrics are defined as follows:

$$Spe = \frac{TN}{TN+FP} \tag{11}$$

$$Pre = \frac{TP}{TP+FP} \tag{12}$$

$$Rec = \frac{TP}{TP+FN} \tag{13}$$

$$F1 = 2 \times \frac{Pre \times Rec}{Pre + Rec} \tag{14}$$

$$Mcc = \frac{TP \times TN - FN \times FP}{\sqrt{(TP+FP) \times (TP+FN) \times (TN+FP) \times (TN+FN)}} \tag{15}$$

Among these metrics, True Positives (TP) refers to the number of samples correctly predicted as positive by the model; False Positives (FP) are negative samples incorrectly predicted as positive; True Negatives (TN) are the number of samples correctly predicted as negative; and False Negatives (FN) are positive samples incorrectly predicted as negative by the model. Specifically, Specificity (Spe) measures the model's ability to correctly identify negative samples; Precision (Pre) reflects the proportion of predicted positive samples that are actually positive; Recall (Rec), also known as Sensitivity, indicates the model's ability to identify all positive samples. Additionally, the F1 score combines the performance of Precision and Recall, while the Matthews Correlation Coefficient (MCC) evaluates the overall performance of the model in handling predictions of both positive and negative classes, particularly suitable for evaluating imbalanced data. Given that this study addresses a binary classification problem with imbalanced classes, MCC is one of our primary evaluation metrics as it provides a comprehensive assessment of such scenarios. A high MCC score is achieved only when the model performs well across all four categories of the confusion matrix (TP, TN, FN, and FP).

## 3. Comparison with existing DNA-binding site predictors

### 3.1 Comparison of iProtDNA-SMOTE with other methods on TE46

To demonstrate the effectiveness of iProtDNA-SMOTE, we compared it against six state-of-the-art models for predicting DNA binding sites:DRNAPred [46], DNAPred, SVMnuc [47], NCBRPred [48], DBPred, and CLAPE-DB [49]. Thes comparisons were based on their performance on TE46 dataset. As detailed in Table 2, iProtDNA-SMOTE outperforms all other methods, with the highest MCCscore. Significantly, iProtDNA-SMOTE surpasses CLAPE-DB, the next best model, by approximately 1.7% in MCC. It also excels across all evaluation metrics. On the TE46 dataset, iProtDNA-SMOTE trained on TR646 achieves specificity (Spe) of 0.973, precision (Pre) of 0.583, F1 score of 0.447, and MCC of 0.418, marking improvements of 13.8%, 27.7%, 1.3%, and 1.7%, respectively, over CLAPE-DB. Although the recall (Rec) of 0.363 is slightly lower than CLAPE-DB, this reflects iProtDNA-SMOTE's emphasis on precision during predictions, effectively reducing false positives. Furthermore, iProtDNA-SMOTE's AUC metric is closely aligned with CLAPE-DB, further substantiating its competitive overall predictive performance.

### 3.2 Comparison of iProtDNA-SMOTE with other methods on TE129 and TE181

Table 3 summarises the performance comparison of various models, including DRNAPred, DNAPred, SVMnuc NCBRPred, CLAPE-DB and iProtDNA-SMOTE on the independent validation dataset TE129. Among these models, iProtDNA-SMOTE achieving the highest MCC score. On the TE129 dataset, iProtDNA-SMOTE, trained with TR573 dataset achieves a specificity of 0.972, precision of 0.497, F1 score of 0.468, MCC of 0.437, and AUC of 0.896. These results represent substantial improvements over CLAPE-DB, with increases of 1.7%, 10.1%, 4.1%, 4.8%, and 1.5%, in specificity, precision, F1 score, MCC, and AUC, respectively.

**Table 2. Performance comparisons of iProtDNA-SMOTE and 6 competing predictors on TE46 under independent validation.**

| Method | Spe | Rec | Pre | F1 | MCC | AUC |
|---|---|---|---|---|---|---|
| DRNAPred | 0.692 | 0.677 | 0.185 | 0.291 | 0.226 | 0.755 |
| DNAPred | 0.655 | 0.671 | 0.157 | 0.254 | 0.194 | 0.730 |
| SVMnuc | 0.666 | 0.668 | 0.154 | 0.250 | 0.192 | 0.715 |
| NCBRPred | 0.674 | 0.677 | 0.165 | 0.265 | 0.207 | 0.713 |
| DBPred | 0.784 | 0.708 | 0.243 | 0.362 | 0.320 | 0.794 |
| CLAPE-DB | 0.835 | **0.747** | 0.306 | 0.434 | 0.401 | **0.871** |
| iProtDNA-SMOTE | **0.973** | 0.363 | **0.583** | **0.447** | **0.418** | 0.850 |

**Table 3. Performance comparisons of iProtDNA-SMOTE and 5 competing predictors on TE129 under independent validation.**

| Method | Spe | Rec | Pre | F1 | MCC | AUC |
|---|---|---|---|---|---|---|
| DRNAPred | 0.937 | 0.233 | 0.190 | 0.210 | 0.155 | 0.693 |
| DNAPred | 0.954 | 0.396 | 0.353 | 0.373 | 0.332 | 0.845 |
| SVMnuc | 0.966 | 0.316 | 0.371 | 0.341 | 0.304 | 0.812 |
| NCBRPred | 0.969 | 0.312 | 0.392 | 0.347 | 0.313 | 0.823 |
| CLAPE-DB | 0.955 | **0.464** | 0.396 | 0.427 | 0.389 | 0.881 |
| iProtDNA-SMOTE | **0.972** | 0.442 | **0.497** | **0.468** | **0.437** | **0.896** |

Table 4 compares the performance of DNAPred, SVMnuc, NCBRPred, CLAPE-DB, and iProtDNA-SMOTE on the TE181 test dataset. iProtDNA-SMOTE achieves the highest MCC value among all methods. On the TE181 dataset, iProtDNA-SMOTE, trained with TR573 dataset, achieves a specificity of 0.963, precision of 0.303, F1 score of 0.330, MCC of 0.299, and AUC of 0.858. These results represent notable improvements over CLAPE-DB, with increases of 3.2% in specificity, 9.1% in precision, 5.0% in the F1 score, 4.7% in MCC, and 3.4% in AUC.

On both the TE129 and TE181 independent test sets, iProtDNA-SMOTE demonstrates recall rates that are nearly on par with CLAPE-DB. This similarity suggests that our model offers a balanced approach to predictions, maintaining high accuracy while carefully avoiding false positives. The close performance in recall between the two models is particularly significant given that CLAPE-DB incorporates contrastive learning and pre-trained protein language models, which are also key components of iProtDNA-SMOTE's deep learning architecture. This comparison underscores the effectiveness of iProtDNA-SMOTE's graph neural network integration and its strategies for tackling class imbalance.

## 4. Conclusions

We introduce iProtDNA-SMOTE, a novel deep learning-based method for predicting DNA binding sites from protein sequences. This approach integrates the pre-trained protein language model ESM2 with graph neural network technology. After through evaluation using five benchmark datasets for protein-DNA binding sites, iProtDNA-SMOTE has been shown to surpass existing state-of-the-art methods in predictive accuracy. Several key advancements contribute to the improvements of iProtDNA-SMOTE. Firstly, the ESM2 model effectively captures the intricate protein sequence features through high-dimensional feature embeddings. Secondly, our graph data augmentation strategy adeptly strengthens the model's capability to identify minority class nodes, leading to enhanced predictive accuracy.

While iProtDNA-SMOTE has demonstrated impressive results, there are opportunities for further refinement. For instance, the current graph model may struggle with extremely long protein sequences, and integrating more sophisticated graph convolutional networks or attention mechanisms could offer improved solutions. Additionally, with the rapid development of protein structure prediction tools such as AlphaFold3 and ESM2, utilizing their predictions could potentially yield even greater accuracy in DNA binding site prediction. Relevant research in these areas is ongoing.

**Table 4. Performance comparisons of iProtDNA-SMOTE and 4 competing predictors on TE181 under independent validation.**

| Method | Spe | Rec | Pre | F1 | MCC | AUC |
|---|---|---|---|---|---|---|
| DNAPred | 0.948 | 0.334 | 0.223 | 0.267 | 0.233 | 0.802 |
| SVMnuc | 0.960 | 0.289 | 0.242 | 0.263 | 0.229 | 0.803 |
| NCBRPred | 0.964 | 0.259 | 0.241 | 0.250 | 0.215 | 0.771 |
| CLAPE-DB | 0.931 | **0.413** | 0.212 | 0.280 | 0.252 | 0.824 |
| iProtDNA-SMOTE | **0.963** | 0.362 | **0.303** | **0.330** | **0.299** | **0.858** |

## Supporting Information

**S1 Tables. Supplementary Tables.**
(DOCX)

**S1 Dataset. iProtDNA-SMOTE benchmark datasets.**
(RAR)

**S1 Code. iProtDNA-SMOTE code.**
(RAR)

**S1 Weight. iProtDNA-SMOTE trained weights.**
(RAR)

**S1 Model. The graph model for dataset TE46.**
(RAR)

## Author contributions

**Conceptualization:** Weizhong Lin.

**Funding acquisition:** Wangren Qiu, Xuan Xiao, Weizhong Lin.

**Methodology:** Weizhong Lin.

**Resources:** Wangren Qiu, Weizhong Lin.

**Software:** Ruiyan Huang.

**Supervision:** Xuan Xiao, Weizhong Lin.

**Writing – original draft:** Ruiyan Huang.

**Writing – review & editing:** Weizhong Lin.

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
