## [Decision Letter · Decision Letter 0]

5 Jan 2025

PONE-D-24-57420iProtDNA-SMOTE: Enhancing Protein-DNA Binding Sites Prediction through Imbalanced Graph Neural NetworksPLOS ONE

Dear Dr. Lin,

Thank you for submitting your manuscript to PLOS ONE. After careful consideration, we feel that it has merit but does not fully meet PLOS ONE’s publication criteria as it currently stands. Therefore, we invite you to submit a revised version of the manuscript that addresses the points raised during the review process.

We look forward to receiving your revised manuscript.

Kind regards,

Syed Nisar Hussain Bukhari

Academic Editor

PLOS ONE

Journal Requirements:

3. Thank you for stating the following financial disclosure: This research was funded by the National Natural Science Foundation of China, 62162032 and 32260154, and Technology Projects of the Education Department of Jiangxi Province of China, GJJ2201040 and GJJ2201004.

4. Thank you for stating the following in the Acknowledgments Section of your manuscript: This research was funded by the National Natural Science Foundation of China, 62162032 and 32260154, and Technology Projects of the Education Department of Jiangxi Province of China, GJJ2201040 and GJJ2201004.

Please remove any funding-related text from the manuscript and let us know how you would like to update your Funding Statement. Currently, your Funding Statement reads as follows: This research was funded by the National Natural Science Foundation of China, 62162032 and 32260154, and Technology Projects of the Education Department of Jiangxi Province of China, GJJ2201040 and GJJ2201004.

5. Your abstract cannot contain citations. Please only include citations in the body text of the manuscript, and ensure that they remain in ascending numerical order on first mention.

Reviewers' comments:

Reviewer's Responses to Questions

**Comments to the Author**

1. Is the manuscript technically sound, and do the data support the conclusions?

Reviewer #1: Yes

Reviewer #2: Yes

Reviewer #3: Yes

Reviewer #4: Yes

Reviewer #5: Partly

2. Has the statistical analysis been performed appropriately and rigorously? 

Reviewer #1: Yes

Reviewer #2: Yes

Reviewer #3: N/A

Reviewer #4: No

Reviewer #5: Yes

3. Have the authors made all data underlying the findings in their manuscript fully available?

Reviewer #1: Yes

Reviewer #2: Yes

Reviewer #3: Yes

Reviewer #4: Yes

Reviewer #5: Yes

4. Is the manuscript presented in an intelligible fashion and written in standard English?

Reviewer #1: Yes

Reviewer #2: Yes

Reviewer #3: Yes

Reviewer #4: Yes

Reviewer #5: Yes

5. Review Comments to the Author

Reviewer #1: 1. The manuscript studies an important area in understanding the biological process and the cellular functions arising out of these interactions.

2. The manuscript is well written, the literature is thoroughly reviewed and the study has been organized in a systematic fashion.

3. The language of the manuscript is good, but needs a little proof reading to fix some language and grammatical errors.

4. The working of ESM2 and Graph SMOTE should have been elaborated within the manuscript, so that it becomes easy for the reader to understand the class balancing and the embeddings generated by ESM2. Although the raw data files on GitHub contain sequence and encoding, but the graph data and ESM2 embedding are in binary format which is beyond comprehension. It would be beneficial for this study to explain the output of ESM2 and the graph structure derived from such embeddings.

5. The authors are also advised to perform some downstream analysis for the novel predictions generated by their model if any to show its relevance in predicting biological functions associated with this DNA binding protein.

Reviewer #2: Considering the use of graph-based neural network structure, it is necessary to discuss and examine more research studies. Also, with further explanations about the innovation presented in the article, the strengths of the presented model can be strengthened.

Reviewer #3: he study is methodologically sound, innovative, and impactful. Addressing the identified weaknesses would further elevate its contributions to the field.

Recommendation: Accept with minor revisions.

Reviewer #4: I find the idea of using graph neural networks and SMOTE to predict protein-DNA binding sites quite intriguing. The experiments on the TR646, TE46, and TR573 datasets, and the comparisons to strong baselines like CLAPE-DB and DNAPred, show promising results with AUC values between 0.850 and 0.896. However, I think the current version needs some serious work before it's ready for a top-tier journal like PLOS ONE.

The first thing that struck me was the huge gap between the method section and the data visualization. The method section felt like a dense wall of text, making it hard to follow. More diagrams or figures to illustrate the model and the results would make it much easier to understand.

I was also disappointed by the lack of discussion about the model's limitations. The authors briefly mention potential issues with long sequences, but that's it. I'd really like to see a more in-depth analysis of things like computational cost, training time, and how well the model scales to larger datasets. This would give a more balanced perspective.

The writing style also felt a bit… robotic. It looks a bit too polished and maybe even a bit salesy. I think a simpler, more direct writing style would be much better.

From a technical standpoint, I was concerned about the lack of ablation studies. The model combines several components, like the ESM2 pre-trained model and GraphSMOTE. It would be really helpful to see how much each of these components actually contributes to the final performance.

Reproducibility is another key issue. The authors provide code and datasets, which is good, but they're missing crucial training details like learning rates, batch sizes, and the number of epochs. This makes it hard for other researchers to independently verify the results.

Finally, the paper doesn't fully address the impact of data imbalance. Even with GraphSMOTE, the recall on the TE46 dataset is quite low (0.363), suggesting that this remains a challenge. I think a deeper discussion on how imbalance affects performance, especially recall, is needed.

Overall, I think the approach has a lot of potential. But the paper needs some significant revisions to make it more readable, transparent, and convincing. I recommend restructuring the paper, simplifying the language, adding more visuals, and conducting more experiments to fully evaluate the model.

Reviewer #5: This paper introduces iProtDNA-SMOTE, a novel model for predicting protein-DNA binding sites. The proposed method addresses the significant class imbalance problem in such datasets by combining the Graph SMOTE algorithm (designed for class imbalance issues) with protein-DNA language models and Graph Neural Networks (GNNs). The model was trained and tested on five protein-DNA binding benchmarks from the literature and demonstrates superior performance compared to other existing models on the same benchmarks.

Given the large class imbalance between the number of residues that bind to DNA and those that do not, the use of the SMOTE algorithm to account for this imbalance is highly relevant. The authors tackle this problem by framing it within a graph-based framework, utilizing embeddings from the ESM model and constructing a graph based on pairwise distances computed from the AlphaFold 3 (AF3) protein structure. The authors then train the Graph Neural Network on datasets curated from prior publications. It is worth noting that these training and test datasets are themselves predictions of protein-DNA interactions derived from previous models (GraphBind, GraphPred, and DBPred). During training, the Graph SMOTE algorithm is employed to upsample examples from the minority class (DNA-binding residues).

Comments:

I find the overall approach of the paper compelling, and it is reasonable to assume that a SMOTE-type algorithm would be beneficial in addressing class imbalance. The results on their independent benchmarks appear promising compared to other models in the literature. Overall, this is an interesting and innovative approach to a biologically significant problem characterized by substantial class imbalance.

However, I would like the following questions addressed before publication:

1. Why are the three models—GraphBind, GraphPred, and DBPred—not included in the benchmarks? The paper does not explain their absence. Is it because their predictions on these benchmarks are already very high, given that the benchmarks (labels) are essentially derived from the predictions of these models? This needs to be clarified, and their performances should be reported, possibly in a supplementary table if necessary.

2. I observed that iProtDNA-SMOTE consistently achieves very high precision but often has the lowest recall across benchmarks. Could the authors address why this trade-off occurs systematically? Is it due to the problem setup of oversampling the minority class, which might make the model adept at identifying a specific type of positive example (protein-DNA binding) while missing others? Some insights or discussion on this issue are crucial. I recommend examining the worst mistakes in the false negatives (i.e., binding sites missed by the model) to better understand the underlying reasons for the low recall and potentially improve it.

6. PLOS authors have the option to publish the peer review history of their article (what does this mean? ). If published, this will include your full peer review and any attached files.

**Do you want your identity to be public for this peer review?** For information about this choice, including consent withdrawal, please see our Privacy Policy .

Reviewer #1: **Yes: ** Nisar Iqbal Wani PhD

Reviewer #2: No

Reviewer #3: **Yes: ** Dr. Syed Mutahar Aaqib

Reviewer #4: No

Reviewer #5: **Yes: ** Abhimanyu Banerjee

---

## [Author Response · Author response to Decision Letter 1]

30 Jan 2025

Dear Editor,

Thank you very much for your Jan-06-2025 email. We appreciate the time and effort that you and the reviewers dedicated to providing feedback on our manuscript. And we are grateful for the insightful and helpful comments on our paper. As suggested, the MS has been carefully revised according to their comments. Our point-to-point responses can be summarized as follows. For clarity, our responses are started with "Reply".

Journal Requirements:

Reply: We have carefully checked and ensured that our manuscript complies with the formatting requirements of PLOS ONE. We have referenced the templates provided by PLOS ONE and made necessary adjustments to the format of our manuscript.

Reply: In accordance with PLOS ONE's guidelines for code sharing, we have made the author-generated code publicly available. The code is accessible at https://github.com/primrosehry/iProtDNA-SMOTE and includes detailed instructions for running it along with dependency information.

3. Thank you for stating the following financial disclosure: This research was funded by the National Natural Science Foundation of China, 62162032 and 32260154, and Technology Projects of the Education Department of Jiangxi Province of China, GJJ2201040 and GJJ2201004.

Reply: We have clearly stated the funding sources in the Funding Statement and added the following declaration: “The funders had no role in study design, data collection and analysis, decision to publish, or preparation of the manuscript.” Additionally, we have removed all funding-related information from the Acknowledgments section to comply with the journal's requirements.

4. Thank you for stating the following in the Acknowledgments Section of your manuscript: This research was funded by the National Natural Science Foundation of China, 62162032 and 32260154, and Technology Projects of the Education Department of Jiangxi Province of China, GJJ2201040 and GJJ2201004.

Reply: We note that you have provided funding information that is not currently declared in your Funding Statement. However, funding information should not appear in the Acknowledgments section or other areas of your manuscript. We will only publish funding information present in the Funding Statement section of the online submission form.

Please remove any funding-related text from the manuscript and let us know how you would like to update your Funding Statement. Currently, your Funding Statement reads as follows: This research was funded by the National Natural Science Foundation of China, 62162032 and 32260154, and Technology Projects of the Education Department of Jiangxi Province of China, GJJ2201040 and GJJ2201004.

Reply: We have removed all funding-related information from the Acknowledgments section and ensured that all relevant declarations appear only in the Funding Statement. We have updated the Funding Statement to ensure its accuracy.

5. Your abstract cannot contain citations. Please only include citations in the body text of the manuscript, and ensure that they remain in ascending numerical order on first mention.

Our abstract does not contain any citations.

Reply: We have ensured that citations appear only in the body of the manuscript and are numbered in ascending order upon their first mention.

Reviewer #1

1. The manuscript studies an important area in understanding the biological process and the cellular functions arising out of these interactions.

Reply: We appreciate your time and effort in evaluating our work. We have expanded the introduction to better emphasize the importance of understanding the biological processes and cellular functions resulting from these interactions. This includes referencing recent studies to highlight the significance of this field.

2. The manuscript is well written, the literature is thoroughly reviewed and the study has been organized in a systematic fashion.

Reply: We appreciate your positive feedback on the manuscript’s writing and organization. We have ensured that the structure remains clear and logical throughout the study.

3. The language of the manuscript is good, but needs a little proof reading to fix some language and grammatical errors.

Reply: This is a good suggestion. We have carefully proofread the manuscript to correct any language and grammatical errors. We have also enlisted the help of a professional language editor to ensure that the manuscript meets high standards of clarity and accuracy.

4. The working of ESM2 and Graph SMOTE should have been elaborated within the manuscript, so that it becomes easy for the reader to understand the class balancing and the embeddings generated by ESM2. Although the raw data files on GitHub contain sequence and encoding, but the graph data and ESM2 embedding are in binary format which is beyond comprehension. It would be beneficial for this study to explain the output of ESM2 and the graph structure derived from such embeddings.

Reply: Thank you for your valuable suggestion. We have addressed this by adding a detail explanation of the workings of ESM2 and Graph SMOTE in manuscript. In section "Unsupervised Protein Language Model" of the Materials and Methods, we have elaborated on how ESM2 works and generates embeddings. Additionally, we have introduced a new section, "Construction of a Balanced Protein Graph," which provides a comprehensive explanation of how Graph SMOTE functions and how these embeddings are used to create graph structures. Furthermore, we have included examples of ESM2 embeddings (Fig. 2) and graphical data (Fig. 3) in a more reader-friendly format for readers.

5. The authors are also advised to perform some downstream analysis for the novel predictions generated by their model if any to show its relevance in predicting biological functions associated with this DNA binding protein.

Reply:We completely agree with your insightful suggestion. While we agree that such analysis would be highly beneficial, we currently face limitations in experimental time and resources that prevent us from conducting the relevant downstream experiments at this stage.

To address this limitation, we have provided a detailed description of the model-building process and the reliability of the prediction results in the manuscript. Our model has been trained and validated on a large dataset of known protein-DNA interaction data, achieving high accuracy and strong generalization capabilities. This rigorous validation process ensures that our model serves as a reliable tool for predicting protein-DNA binding sites.

We believe that these revisions have significantly improved the manuscript and addressed your concerns. We are grateful for your suggestions and hope that the revised version meets your expectations.

Reviewer #2

Considering the use of graph-based neural network structure, it is necessary to discuss and examine more research studies. Also, with further explanations about the innovation presented in the article, the strengths of the presented model can be strengthened.

Reply: Thank you for your valuable feedback on our manuscript. We highly appreciate Reviewer#2’s suggestion.

1- Given the main structure of the paper, which is based on graph-based neural networks, more previous research needs to be studied.

Reply: Many thanks for the reviewer’s suggestion. We have expanded our literature review to include additional studies on graph-based neural networks. The expansion provides a more comprehensive overview of the field.

2- In the classification of Graph structured data section, graph convolution operations need to be discussed further.

Reply:This is a good point. We have added a detailed explanation of the graph convolution operations in the "GraphSAGE-MLP Network" section, clarifying their role in feature aggregation.

3- Also in the classification of Graph structured data section, more explanation should be provided about collecting neighbor features.

Reply: We think this is an excellent suggestion. In the "GraphSAGE-MLP Network" section, we have provided a more comprehensive explanation of how neighbor features are collected, with a focus on the message-passing mechanism and its implementation in the model.

4- In Table 1, the training and test datasets are different. Please explain why this is done.

Reply: Many thanks for the reviewer’s suggestion. In Table 1, we have used two datasets for training and testing. One dataset comprises the training set TR646 and independent test set TE46, while the another dataset includes the training set TR573 and the independent test sets TE129 and TE181.

In the field of protein-DNA binding site prediction, these five classic datasets are widely used for model training and testing. The separation of the training set and the test set is essential to ensure the model's generalization ability. The training set is used to enable the model to learn the features of protein-DNA interactions, while the independent test set is used to evaluate the model's performance on unseen data. This separation helps to prevent overfitting and ensures the reliability and objectivity of the results.

Reviewer #3

The study is methodologically sound, innovative, and impactful. Addressing the identified weaknesses would further elevate its contributions to the field.

Recommendation: Accept with minor revisions.

Reply: We highly appreciate your positive comments and encouragement.

1. Sensitivity Analysis:

Include experiments to analyze the trade-offs between precision and recall for various datasets, particularly focusing on the biological implications of missing DNA-binding residues.

Reply: We appreciate the reviewer’s valuable suggestion. We have conducted a detailed analysis of the trade-offs between precision and recall for various datasets, particularly focusing on the biological implications of missing DNA-binding residues. This analysis is included in the "Results" and "Conclusions" sections of our manuscript. We have also outlined potential future research directions aimed at significantly improving recall while maintaining high precision, which will further enhance the overall performance of our model.

2. Efficiency Metrics:

Provide a comparison of computational time and resource utilization against competing methods to offer a holistic evaluation of the model’s practicality.

Reply: This is a good suggestion. In the "Conclusions" section, we have added a discussion on the computational resources used during our study to reduce computational time. We have included key training details such as dropout, alpha, gamma, learning rate, and epochs in section "Results" to enhance the reproducibility of our study. This information will help other researchers more accurately replicate our experimental results and compare computational efficiency.

3. Future Directions:

Discuss potential integrations with advanced graph attention mechanisms or hybrid models to address current limitations in handling long protein sequences.

Consider incorporating more diverse datasets or synthetic benchmarks to evaluate robustness further.

Reply: We thank the reviewer for pointing out this issue. In the "Conclusions" section, we have added a discussion on the limitations of the model and proposed potential directions for future research to address these limitations. Specifically, we highlighted the need to further optimize the model's prediction strategy to improve recall while maintaining high precision. We plan to introduce more complex graph convolutional network architectures, integrate protein structure prediction tools, and adjust the model's prediction threshold. These improvements are expected to enhance the overall performance of the model.

4. Error Analysis:

A deeper error analysis to identify specific cases where the model underperforms (e.g., specific protein classes or sequence patterns) would provide actionable insights for further refinement.

Reply: We agree with the reviewer’s suggestion. In the final part of section "Results," we have added an analysis of the impact of GraphSMOTE on model performance, conducting a more in-depth examination of how data imbalance affects model performance, particularly recall. Although our model's recall (Rec) value is lower than that of CLAPE-DB, it outperforms CLAPE-DB in terms of precision (Pre) and other performance metrics. This reflects iProtDNA-SMOTE's emphasis on precision during the prediction process, effectively reducing false positives. In the "Conclusions" section, we have also added a discussion on the limitations of the model and proposed potential directions for future research to address these limitations.

Reviewer #4

I find the idea of using graph neural networks and SMOTE to predict protein-DNA binding sites quite intriguing. The experiments on the TR646, TE46, and TR573 datasets, and the comparisons to strong baselines like CLAPE-DB and DNAPred, show promising results with AUC values between 0.850 and 0.896. However, I think the current version needs some serious work before it's ready for a top-tier journal like PLOS ONE.

Reply: We deeply appreciate Reviewer#4’s overall positive feedback and constructive comments.

The first thing that struck me was the huge gap between the method section and the data visualization. The method section felt like a dense wall of text, making it hard to follow. More diagrams or figures to illustrate the model and the results would make it much easier to understand.

Reply: We completely agree with this valuable suggestion by the reviewer. In response to your suggestion, we have added three new figures (see in Fig. 2, Fig. 3, and Fig. 4) to illustrate the key aspects of our model.

I was also disappointed by the lack of discussion about the model's limitations. The authors briefly mention potential issues with long sequences, but that's it. I'd really like to see a more in-depth analysis of things like computational cost, training time, and how well the model scales to larger datasets. This would give a more balanced perspective.

Reply: This is a good suggestion. In the "Conclusions" section, we have added a discussion on the limitations of the model and proposed potential directions for future research to address these limitations.

The writing style also felt a bit… robotic. It looks a bit too polished and maybe even a bit salesy. I think a simpler, more direct writing style would be much better.

Reply: We thank the reviewer for highlighting this issue. In response, we have simplified the language and made the writing more direct and accessible. We hope this improves the readability and clarity of our manuscript.

From a technical standpoint, I was concerned about the lack of ablation studies. The model combines several components, like the ESM2 pre-trained model and GraphSMOTE. It would be really helpful to see how much each of these components actually contributes to the final performa

---

## [Decision Letter · Decision Letter 1]

25 Feb 2025

iProtDNA-SMOTE: Enhancing Protein-DNA Binding Sites Prediction through Imbalanced Graph Neural Networks

PONE-D-24-57420R1

Dear Dr. Lin,

We’re pleased to inform you that your manuscript has been judged scientifically suitable for publication and will be formally accepted for publication once it meets all outstanding technical requirements.

Kind regards,

Syed Nisar Hussain Bukhari

Academic Editor

PLOS ONE

Additional Editor Comments (optional):

Reviewers' comments:

Reviewer's Responses to Questions

**Comments to the Author**

1. If the authors have adequately addressed your comments raised in a previous round of review and you feel that this manuscript is now acceptable for publication, you may indicate that here to bypass the “Comments to the Author” section, enter your conflict of interest statement in the “Confidential to Editor” section, and submit your "Accept" recommendation.

Reviewer #2: (No Response)

2. Is the manuscript technically sound, and do the data support the conclusions?

Reviewer #2: (No Response)

3. Has the statistical analysis been performed appropriately and rigorously? 

Reviewer #2: (No Response)

4. Have the authors made all data underlying the findings in their manuscript fully available?

Reviewer #2: (No Response)

5. Is the manuscript presented in an intelligible fashion and written in standard English?

Reviewer #2: (No Response)

6. Review Comments to the Author

Reviewer #2: (No Response)

7. PLOS authors have the option to publish the peer review history of their article (what does this mean? ). If published, this will include your full peer review and any attached files.

**Do you want your identity to be public for this peer review?** For information about this choice, including consent withdrawal, please see our Privacy Policy .

Reviewer #2: No

---

## [Editor Report · Acceptance letter]

PONE-D-24-57420R1

PLOS ONE

Dear Dr. Lin,

I'm pleased to inform you that your manuscript has been deemed suitable for publication in PLOS ONE. Congratulations! Your manuscript is now being handed over to our production team.

Kind regards,

on behalf of

Dr. Syed Nisar Hussain Bukhari

Academic Editor

PLOS ONE